# The Role of Pharmacists in Providing Pharmaceutical Care in Primary and Secondary Prevention of Stroke: A Systematic Review and Meta-Analysis

**DOI:** 10.3390/healthcare10112315

**Published:** 2022-11-18

**Authors:** Saeed Al-Qahtani, Zahraa Jalal, Vibhu Paudyal, Sajid Mahmood, Julie Mason

**Affiliations:** 1School of Pharmacy, Institute of Clinical Sciences, University of Birmingham, Birmingham B15 2TT, UK; 2School of Pharmacy, Jazan University, Jazan 45142, Saudi Arabia; 3Department of Pharmacy, Quaid-i-Azam University, Islamabad 45320, Pakistan

**Keywords:** primary management, pharmacist role, secondary management, stroke

## Abstract

Pharmacists deliver pharmaceutical care in many different healthcare settings and are well-placed to support the prevention of stroke. However, their role and impact in this area is ill-defined. This systematic review aims to explore the pharmacists’ role in stroke prevention. Nine databases were searched for studies reporting pharmacist interventions in the management of primary and secondary ischaemic stroke prevention. Study quality was evaluated through Cochrane Risk of Bias and Joanna Briggs Institute (JBI) appraisal tools where possible. A narrative review was conducted and meta-analysis performed for studies with comparable outcomes. Of the 834 initial articles, 31 met inclusion criteria. Study designs were varied and included controlled trials, observational studies, audit reports and conference abstracts. Seven studies addressed the pharmacists’ role in primary prevention and 24 in secondary prevention. Pharmacist interventions reported were diverse and often multifactorial. Overall, 20 studies reported significant improvement in outcomes. Meta-analysis showed pharmacist interventions in emergency care significantly improved the odds of achieving thrombolytic therapy door to needle (DTN) times ≤45 min, odds ratio: 2.69 (95% confidence interval (CI): 1.95–3.72); *p* < 0.001. The pharmacists’ role is varied and spans the stroke treatment pathway, with the potential for a positive impact on a range of health-related outcomes.

## 1. Introduction

Stroke is considered as a major cause of mortality and disability, and can be associated with significant economic cost [1]. Worldwide, death rates associated with stroke are decreasing, yet prevalence rates are increasing. In developing countries, stroke is still reported as the second most common cause of mortality [2]. After a first stroke, it is estimated that 11% of individuals will have a recurrence within a year and 26% within 5 years [3]. Evidence suggests that secondary prevention through the management of risk factors, such as hypertension, dyslipidaemia and the use of antiplatelet treatment can reduce the risk of stroke recurrence by up to 30% [4,5,6].

With an aging global population, the importance of effective interventions by healthcare professionals to reduce stroke risk and improve treatment outcomes is well recognised [7]. While primary prevention interventions have the potential to reduce the risk of stroke in asymptomatic people, secondary prevention interventions potentially reduce the risk of recurrence [8]. Pharmacists are one of the most accessible healthcare professions. Their presence in high-street, community-based premises allows convenient public access to health advice without appointment. This offers a great potential for raising awareness through educational health campaigns and for preventative healthcare through screening services. Patients may also encounter pharmacists in hospitals and increasingly in primary care general practitioner surgeries. As experts in medicines and the management of medicines and given the variety of healthcare settings that pharmacists work within, they have the potential to play a vital role in the primary and secondary prevention of stroke.

Initiatives such as the UK’s community pharmacy New Medicine Service (NMS), which provides support to patients with chronic conditions to improve medication adherence, offer opportunities for pharmacists to support and educate patients with chronic conditions such as diabetes, hypertension, and stroke [9]. In addition to the community pharmacy, pharmacist-led medication management in many settings (e.g., hospitals, general practice, and outpatient clinics) can improve health outcomes, improve modifiable risk factors, minimize healthcare system costs, enhance medication safety and improve patient satisfaction [10,11,12,13,14].

Literature providing an overview of the pharmacist role in both primary and secondary prevention of stroke is limited. Previous systematic reviews have concentrated on pharmacist interventions for either the primary or secondary prevention of stroke in specific settings/work environments. This systematic review and meta-analysis aim to explore where and how pharmacists and the services that they provide can impact and improve outcomes for the primary and secondary prevention of stroke.

## 2. Methods

This review was guided by the Cochrane Handbook for Systematic Reviews of Interventions [15] and reported in line with the Preferred Reporting Items for Systematic Reviews and Meta-Analyses statement [16]. The review protocol was registered with the International prospective register of systematic reviews (PROSPERO ID: CRD42019151267).

### 2.1. Search Strategy

A systematic search for peer-reviewed healthcare-related articles was performed using the following databases: PubMed, EMBASE, MEDLINE, Google Scholar, PsycINFO, CINAHL Plus, SCOPUS, the Science Citation Index, and the Cochrane Library (1974 to December 2021). The search strategy employed four key themes: stroke, pharmacist’s role, primary prevention, and secondary prevention. Stroke-related search and MeSH terms included: stroke, cerebrovascular accident, brain ischaemia, and ischaemic stroke. Terms associated with the pharmacist’s role included: pharmacists, pharmacist’s role, pharmaceutical care, clinical/hospital and community pharmacists, advice, counsel*, advis*, and educat*. For prevention, the MeSH terms primary prevention and secondary prevention were used. All searches were restricted to English language articles only.

### 2.2. Study Selection

Search results were imported into and managed within EndNote 8.1 software. Duplicates were removed and one author (SA) identified potentially relevant articles through title and abstract screening. Full texts of potentially relevant articles were then retrieved where available. One author (SA) reviewed full texts for eligibility. Where ambiguities were identified, full texts were independently reviewed for eligibility by two authors (ZJ and JM). Where consensus could not be reached another author (VP) was consulted and discrepancies resolved by discussion. Studies were deemed relevant if they included: interventions by pharmacists or pharmacy students in the management of primary and secondary prevention of stroke; interventions to improve medication adherence; evaluation of pharmacist interventions and clinical care outcomes in stroke patients; studies delivered in any setting where the role of the pharmacist was highlighted in delivering pharmaceutical care to stroke patients. Studies involving haemorrhagic stroke or ischaemic stroke in children were excluded due low incidence and different management practices [17]. Figure 1 shows the PRISMA flow chart for the process.

### 2.3. Data Extraction

Included articles were categorised by how they reported the pharmacist’s role in either the primary or secondary prevention of stroke. Secondary prevention was further sub-categorised into emergency and acute treatment of stroke and long-term management. Data extraction was conducted by the main author (SA) and checked by a second researcher (JM). The data extracted from each eligible study was entered into a specially designed Microsoft Excel (2108) spreadsheet data collection form. Data gathered included: author and year of publication, target population, study aim, study design and setting, description of pharmacist/pharmacy student intervention, and result/outcome.

### 2.4. Data Analysis

Due to the heterogeneity of research design and outcomes measured, the included studies were divided into either primary stroke prevention or secondary stroke prevention for analysis. Studies categorised as secondary stroke prevention fell into two further sub-categories, acute and long-term management. For studies categorised as primary stroke prevention and secondary prevention, due to long-term management, diversity of study design and outcomes measures only descriptive and narrative synthesis was possible. However, for studies categorised as acute secondary prevention, similarity of clinical outcome measures allowed meta-analysis. Only clinically and statistically homogenous studies were combined for meta-analysis. The clinical outcome used was the percentage difference in Door to Needle times for the administration of thrombolytic in hospitals where a pharmacist was present versus where a pharmacist was absent. Statistical analysis was conducted using the Review Manager, version 5.4 (RevMan 5) computer programme (The Cochrane Collaboration, 2020). The clinical homogeneity was assessed in terms of methods used for assessment of the event and study population. A fixed effect model was used for the estimation of risk ratios. Statistical heterogeneity within pooled studies was tested through I^2^ statistics with a 95% confidence. An I^2^ value of less than 50% was considered to indicate substantial clinical homogeneity [18]. Since the level of heterogeneity among included studies was very low (i.e., 24%), sensitivity analysis was deemed unnecessary.

### 2.5. Quality Assessment

The included studies were grouped by design to allow quality assessment. Randomised Controlled Trials (RCTs) were assessed using the Cochrane Risk of Bias tool [18] to determine the risk of bias associated with seven domains. Using this method, the risk of bias was classified as either high, unclear, or low (Table 1). For all other study designs the appropriate Joanna Briggs Institute (JBI) critical appraisal tools [19] were used; cross-sectional (Table 2), quasi-experimental (Table 3), and cohort (Table 4). Every item in JBI checklists was answered either yes, no, unclear (NC) or not applicable (NA). To allow comparison of quality across the various study designs, the percentage of positive assessments of checklist questions (for JBI assessment tools) or low risk of bias domains (for Cochrane assessment tools) was calculated for each of the included studies. Quality ranking was allocated as low (less than 33%), medium (33–66%) or high (over 66%) [20].

## 3. Results

The literature search identified 834 articles of which 31 met the inclusion criteria (Figure 1). Study designs were varied: seven were controlled trials, including four randomized by participant [21,22,23,24], two cluster randomized [25,26] and one non-randomized [27]. Of the cluster randomized studies, one [26] was a sub-analysis of the completed study [25]. Three studies were of experimental before-and-after design [28,29,30], five were cross-sectional [31,32,33,34,35] and fourteen were cohort studies [36,37,38,39,40,41,42,43,44,45,46,47,48,49]. To capture the breadth of pharmacist interventions, two audit reports [50,51] and five conference abstracts [23,33,40,41,47] were also included for data extraction although quality assessment was not possible using standard tools. Most of the studies assessed were of medium or high quality. All but one of the included RCTs were assessed as high quality (Table 1). For cross-sectional studies, the quality score ranged from 50% (medium) to 75% (high) (Table 2). All quasi-experimental studies were assessed as high quality with scores of greater than 66% (Table 3) and cohort study quality assessment scores ranged from 54.5% (medium) to 81.8% (high) (Table 4). Individual study characteristics are shown in Table 5, Table 6 and Table 7. Seven studies reported pharmacist interventions for primary stroke prevention [28,29,30,32,33,36,50] and twenty-four focussed on secondary stroke prevention [21,22,23,24,25,26,27,31,34,35,37,38,39,40,41,42,43,45,46,47,48,49,51].

**Table 1 healthcare-10-02315-t001:** Assessment of the Cochrane Risk of Bias for the included Randomized Controlled Trials (RCTs).

Risk of Bias Domain	Chiu et al. [21]	Hedegaard et al. [22]	Hohmann et al. [26]	Hohmann et al. [25]	McAlister et al. [24]
Selection bias (random sequence generation)	**Low**	**Low**	**Low**	**Low**	**Low**
Selection bias (allocation concealment)	**High**	**Low**	**Low**	**High**	**High**
Reporting bias (selective reporting)	**High**	**Low**	**Low**	**Low**	**Low**
Other bias (other sources of bias)	**Low**	**Low**	**Low**	**Low**	**Low**
Performance bias—blinding (participants and personnel)	**High**	**Low**	**Low**	**Low**	**Low**
Detection bias—blinding (outcome assessment)	**High**	**Low**	**Low**	**High**	**High**
Attrition bias (Incomplete outcome data)	**Low**	**Low**	**Low**	**Low**	**Low**
**Total quality assessment score for each study**	**43%** **(Medium)**	**100%** **(High)**	**100%** **(High)**	**71%** **(High)**	**71%** **(High)**

**Table 2 healthcare-10-02315-t002:** Assessment of included cross-sectional studies using Joanna Briggs Institute (JBI) critical appraisal tools (where NC indicates unclear).

JBI Critical Appraisal Checklists for Cross-Sectional Studies	Hohmann et al. [34]	Lindblad and Howorko [31]	Lowres et al. [32]	Pandya et al. [35]
Were the criteria for inclusion in the sample clearly defined?	**YES**	**YES**	**YES**	**YES**
Were the study subjects and the setting described in detail?	**YES**	**YES**	**YES**	**YES**
Was the exposure measured in a valid and reliable way?	**NC**	**YES**	**YES**	**NO**
Were objective, standard criteria used for measurement of the condition?	**YES**	**YES**	**YES**	**YES**
Were confounding factors identified?	**NO**	**NC**	**NC**	**NO**
Were strategies to deal with confounding factors stated?	**NO**	**NC**	**NC**	**NO**
Were the outcomes measured in a valid and reliable way?	**NC**	**NC**	**YES**	**YES**
Was appropriate statistical analysis used?	**YES**	**NC**	**YES**	**NO**
**Total quality assessment score for each study**	**50%** **(Medium)**	**50%** **(Medium)**	**75%** **(High)**	**50%** **(Medium)**

**Table 3 healthcare-10-02315-t003:** Assessment of quasi-experimental studies (i.e., before–after or non-randomized study designs) using Joanna Briggs Institute (JBI) critical appraisal tools.

JBI Critical Appraisal Checklists for Quasi-Experimental Studies	Bajorek et al. [28]	Hohmann et al. [27]	Jackson and Peterson [29]	Vo et al. [30]
Is it clear in the study what is the ‘cause’ and what is the ‘effect’ (i.e., there is no confusion about which variable comes first)?	**YES**	**YES**	**YES**	**YES**
Were the participants included in any comparisons similar?	**YES**	**YES**	**NO**	**YES**
Were the participants included in any comparisons receiving similar treatment/care, other than the exposure or intervention of interest?	**YES**	**NO**	**YES**	**YES**
Was there a control group?	**NO**	**YES**	**NO**	**NO**
Were there multiple measurements of the outcome both pre and post the intervention/exposure?	**YES**	**YES**	**NO**	**YES**
Was follow up complete and if not, were differences between groups in terms of their follow up adequately described and analysed?	**YES**	**YES**	**YES**	**NO**
Were the outcomes of participants included in any comparisons measured in the same way?	**YES**	**YES**	**YES**	**YES**
Were outcomes measured in a reliable way?	**YES**	**YES**	**YES**	**YES**
Was appropriate statistical analysis used?	**YES**	**YES**	**YES**	**YES**
**Total quality assessment score for each study**	**88.9% (High)**	**88.9% (High)**	**66.7% (High)**	**77.8% (High)**

**Table 4 healthcare-10-02315-t004:** Assessment of included cohort studies using Joanna Briggs Institute (JBI) critical appraisal tools (where NC indicates unclear, and NA indicates not applicable).

JBI Critical Appraisal Checklists for Cohort Studies	Andres et al. [45]	Gosser et al. [37]	Greger et al. [49]	Lee et al. [36]	Montgomery et al. [38]	Nathans et al. [39]	Rech et al. [42]
Were the two groups similar and recruited from the same population?	**YES**	**YES**	**YES**	**YES**	**YES**	**YES**	**YES**
Were the exposures measured similarly to assign people to both exposed and unexposed groups?	**YES**	**YES**	**YES**	**YES**	**YES**	**YES**	**YES**
Was the exposure measured in a valid and reliable way?	**YES**	**YES**	**YES**	**YES**	**YES**	**YES**	**YES**
Were confounding factors identified?	**NO**	**NO**	**YES**	**NO**	**NO**	**YES**	**YES**
Were strategies to deal with confounding factors stated?	**NO**	**NO**	**YES**	**NO**	**NO**	**YES**	**NO**
Were the groups/participants free of the outcome at the start of the study (or at the moment of exposure)?	**YES**	**YES**	**YES**	**YES**	**YES**	**YES**	**YES**
Were the outcomes measured in a valid and reliable way?	**YES**	**YES**	**YES**	**YES**	**YES**	**YES**	**YES**
Was the follow up time reported and sufficient to be long enough for outcomes to occur?	**YES**	**NA**	**YES**	**NC**	**NA**	**NC**	**YES**
Was follow up complete, and if not, were the reasons to loss to follow up described and explored?	**YES**	**NA**	**NC**	**NC**	**NA**	**NC**	**NO**
Were strategies to address incomplete follow up utilized?	**NC**	**NA**	**NO**	**NO**	**NA**	**NC**	**NO**
Was appropriate statistical analysis used?	**NO**	**YES**	**YES**	**YES**	**YES**	**YES**	**YES**
**Total quality assessment score for each study**	**63.6% (Medium)**	**54.5% (Medium)**	**81.8% (High)**	**54.5% (Medium)**	**54.5% (Medium)**	**72.7% (High)**	**72.7% (High)**

**Table 5 healthcare-10-02315-t005:** Primary prevention interventions.

Author(s) and Year of Publication	Target Population	Study Aim(s)	Study Design & Setting	Description of Pharmacist/Pharmacy Student Intervention	Results/Outcomes
**Bajorek et al.** [28]	Patients aged ≥65 years, diagnosed with AF or at high risk of AF.	To establish, assess, and perform a multidisciplinary pharmacist-led hospital-based intervention to optimize the antithrombotic therapy in elderly patients with AF at high risk of stroke.	**Before and after study** conducted over 6 months in an Australian teaching hospital.	Pharmacist-led screening, interview, communication, education, consultation, risk assessment and recommendations for suitable antithrombotic therapy for AF patients based on a pre-set algorithm.	78 of 218 patients (35.8%) required changes to their existing antithrombotic therapy. 60 of the 78 therapy changes (76.9%) were to more-effective treatment options. The proportion of patients with AF protected with antithrombotic therapy was significantly increased at discharge from (59.6% pre-intervention to 81.2% post-intervention, ***p* < 0.001**).
**Jackson and Peterson** [29]	Patients diagnosed with AF or at high risk of AF.	To implement and assess a pharmacist-led stroke risk assessment for hospital in-patients with AF.	**Before and after study** conducted over 17 months in an Australian hospital.	Pharmacist-led stroke risk assessment for AF patients and recommendations regarding suitable antithrombotic therapy.	50 of 134 (37%) of patients assessed were recommended a change in therapy; 44 of these recommendations resulted in a change to antithrombotic therapy when compared to admission. 30 of the 44 therapy changes (68%) were to more effective treatment options. The use of warfarin at discharge was significantly increased compared to admission (74% pre-intervention to 98% post-intervention, ***p* < 0.001**).
**Lee et al.** [36]	Patients diagnosed with AF and initially prescribed dabigatran.	To determine if pharmacist monitoring of dabigatran therapy in the first months (3 months) of treatment in patients with AF improves adherence.	**Retrospective cohort study** conducted over 13 months in an American VA hospital.	Pharmacist-led adherence education about dabigatran, and follow-up telephone calls or visits.	No significant difference in adherence as measured by MPR between intervention and groups ACC (*n* = 20), UC (*n* = 48) over 3 months. Mean MPR values in (ACC pharmacist) = 93.1% and UC = 88.3%, (***p* = 0.16**).
**Lowres et al.** [32]	Older adults, aged ≥65 years, attending (customers of) selected community pharmacies.	To find out the impact, utility and cost-effectiveness of screening in community pharmacy by using iPhone ECG technology to identify undiagnosed AF with referral to GP for management and review. The eventual aim was to reduce stroke and thromboembolism burden.	**Cross sectional study** conducted over 8 months in 10 Australian community pharmacies.	Pharmacist screening of medical history, pulse palpation, ECG test and interpretation with GP referral in cases of suspected AF.	1.5% (95% CI, 0.8–2.5%) of 1000 pharmacy customers were newly detected with AF. Prevalence of AF was 6.7%. The cost-effectiveness of ICER per QALY gained and stroke prevention of screening against unscreened women and men at age 65 to 84 years was calculated. ICER for one stroke prevention was $AUD 30,481, and ICER per QALY was $AUD 5988.
**Papastergiou** [33]	Patients at risk of AF or QT-interval prolongation or a CHADS2 score of >2.	To assess iECG screening in community pharmacies for the detection of undiagnosed AF in patients at high risk.	**Cross sectional study** conducted in 2 Canadian community pharmacies. **(Only published abstract available)**	Pharmacist screening and Interpreting (iECG) reading.	10 (28.6%) of the 35 high risk patients were found to have on abnormal rhythms and were referred to their primary care physician.
**Virdee and Stewart** [50]	Patients with AF history and not anticoagulated and with a CHA_2_DS_2_-VASc ≥1/≥2 (*n* = 497).	To evaluate the level of anticoagulation usage in patients with a CHA_2_DS_2_-VASc ≥1/≥2 (male/female) according to NICE guidelines. The role of pharmacist intervention to optimize therapy.	**Clinical audit** against NICE guidelines using 12 months of data from 15 UK medical practices.	Pharmacist review of patient medical records and discussion with GPs for optimisation of anticoagulant therapy.	65.8% (*n* = 327) of patients were not taking anticoagulants in accordance with NICE guidelines. 77% of 382 pharmacist recommendations to optimize therapy were agreed by GPs.
**Vo et al.** [30]	General public (attendees at community health fairs) aged ≥18 years.	To assess the effect of Act FAST educational intervention carried out by pharmacy students on public alertness.	**Before and after study** Community health fairs held over a 10-month period in Vallejo, US.	Pharmacy student general health screening of blood glucose levels and blood pressure with a 10-min educational intervention.	The questionnaire was used to assess the knowledge of public regarding signs, symptoms, management. The scale for assessing the knowledge included low knowledge (<2), moderate knowledge (3), and high knowledge (≥4). The scale for assessing the knowledge regarding the risk factors of stroke included low knowledge (≤4), moderate knowledge (6–8), and high knowledge (9). The scores were measured in both scales by summing the correct answers out of 6 and 9 total potential correct response. Participant knowledge of the signs, symptoms, risk factors and management of stroke improved significantly post-intervention (***p* < 0.0001**).

ACC (Anticoagulation clinic); AF (Atrial Fibrillation); CHADS2 (congestive heart failure, hypertension, age (>65 = 1 point, >75 = 2 points), diabetes, previous stroke/transient ischemic attack); CHA_2_DS_2_-VASc (congestive heart failure, hypertension, age > 75 years (doubled), type 2 diabetes mellitus, previous stroke, transient ischemic attack or thromboembolism (doubled), vascular disease, age of 65–75 years, and sex); CI (Confidence Interval); GP (General Practitioner); ICER (Incremental Cost-Effectiveness Ratio); iECG (iPhone-based lead-I electrocardiography); MPR (Medication Possession Ratio); NICE (National Institute for Health and Care Excellence); QALY (Quality Adjusted Life Year); UC (Usual Care); UK (United Kingdom); US (United States); VA (Veterans Affairs).

**Table 6 healthcare-10-02315-t006:** Secondary prevention interventions for emergency and acute care.

Author(s) and Year of Publication	Target Population	Study Aim(s)	Study Design & Setting	Description of Pharmacist Intervention	Results/Outcomes
Barbour et al. [46]	Patients who received thrombolytic (rtPA) for treatment of acute ischaemic stroke at ED.	To assess the impact of pharmacist presence on DTN times and patient outcomes.	**Retrospective cohort** study conducted in a US stroke centre with data collection covering a period of 3 years and 10 months.	Addition of a pharmacist to the ‘stroke response team’ with responsibility for thrombolytic (rtPA) contraindications, screening dose calculation and preparation.	164 patient records were included with 31 allocated to the pharmacist present group and 133 to the pharmacist absent group. **Median Door-to-needle (rtPA) times:** Pharmacist present median 35 min vs. pharmacist absent median 42 min; (***p* = 0.003**). ≤30 min achieved in 11 of 31 cases (35.5%) when pharmacist present vs. 22 of 133 cases (16.5%) in the pharmacist absent group (***p* = 0.018**). ≤45 min achieved in 25 of 31 cases (80.7%) when pharmacist present vs. 76 of 133 cases (57.1%) in the pharmacist absent group (***p* = 0.015**). **Median NIHSS scores at discharge:** Pharmacist present median score of 2 (IQR 0–5) vs. pharmacist absent median score of 4 (IQR 0.25–8.75); (***p* = 0.049**).
Bayies et al. [47]	Patients who received thrombolytic (rtPA) for treatment of acute ischaemic stroke at ED.	To assess the impact of pharmacist presence on DTN times.	**Retrospective cohort** study conducted in a US stroke centre with data collection covering a period of 1 year and 6 months. **(Only published abstract available)**	NA	**Median Door-to-needle (rtPA) times:** Pharmacist present median 42.5 min vs. pharmacist absent median 58 min. <45 min achieved in 17 of 30 cases (57%) when pharmacist present vs. 7 of 22 cases (32%) in the pharmacist absent group.
Brandon et al. [41]	Patients who received thrombolytic (rtPA) for treatment of acute ischaemic stroke at ED.	To improve the process for thrombolytic dose calculation and preparation to reduce administration times. Establish zero min decision to needle time goal in addition of pharmacist to stroke team.	A cohort study conducted in unnamed US hospital(s) with a retrospective chart review and prospective data collection post intervention. **(Only published abstract available)**	Addition of a pharmacist to the ‘code stroke team’ with responsibility for thrombolytic (rtPA) dose calculation and preparation plus patient/carer counselling and education.	Improved decision to needle times reported. In the year prior to the intervention, the target zero-minute decision to needle time was not achieved (no case numbers reported). Post intervention zero-minute decision to needle time targets were achieved in 60% of administrations and 78% of administrations in the subsequent two years (no case numbers or actual times reported).
Gosser et al. [37]	Patients aged ≥18 years who received thrombolytic (rtPA) for acute ischaemic stroke.	To examine the impact of pharmacist involvement on rtPA dosing accuracy and door-to-needle time.	**Retrospective cohort study** conducted in a US stroke centre with data collection covering a period of 4 years and 9 months. Participants were assigned to a pharmacist present or pharmacist absent group for analysis.	Pharmacist involvement included documentation in notes, order entry and/or dispensing.	105 patient records were included with 67 allocated to the pharmacist present group and 38 to the pharmacist absent group. **Dosing accuracy:** Pharmacist present 96.6% vs. pharmacist absent 95.6% (***p* = 0.8953**). **Door-to-needle (rtPA) times:** Pharmacist present median 69.5 min vs. pharmacist absent median 89.5 min; (***p* = 0.0027**). <60 min achieved in 20 of 67 cases (29.9%) when pharmacist present vs. 6 of 38 cases (15.8%) in the pharmacist absent group (***p* = 0.1087**).
Hosoya et al. [48]	Patients who received thrombolytic (rtPA) for treatment of acute ischaemic stroke at ED.	To evaluate the pharmacists’ role on rtPA therapy for acute ischaemic stroke patients.	**Retrospective cohort study** conducted in a Japanese hospital with data collection covering a period of 4 years.	Investigating medications administration and patient allergies information through patient notebook and patient family and rtPA preparation.	**Median Door-to-needle (rtPA) times:** Pharmacist present median 74 min vs. pharmacist absent median 89 min; (***p* = 0.01**).
Jacoby et al. [43]	Patients who received thrombolytic (rtPA) for treatment of acute ischaemic stroke at ED.	To assess the impact of pharmacist presence on DTN times and patient outcomes.	**Retrospective cohort study** conducted in a US stroke centre with data collection covering a period of 2 years and 1 month.	Interview patients and their family and review medication history, manage patient’s blood glucose and pressure, verifying INR for patients on warfarin, calculate rtPA dose, prepare and administer rtPA.	100 patient records were included with 49 allocated to the pharmacist present group and 51 to the pharmacist absent group. **Median Door-to-needle (rtPA) times:** Pharmacist present median 46 min vs. pharmacist absent median 58 min; (***p* = 0.019**). ≤45 min achieved in 24 of 49 cases (49%) when pharmacist present vs. 13 of 51 cases (25%) in the pharmacist absent group (*p* = 0.015). ≤60 min achieved in 35 of 49 cases (71%) when pharmacist present vs. 31 of 51 cases (61%) in the pharmacist absent group (*p* = 0.261). **Median NIHSS scores at 24 h post-rtPA:** Pharmacist present median score of 1 (IQR 0–4) in 43 cases vs. pharmacist absent median score of 2 (IQR 0–9.25) in 46 cases; (*p* = 0.047) **Median NIHSS scores at discharge:** Pharmacist present median score of 0 (IQR 0–4) in 44 cases vs. pharmacist absent median score of 2 (IQR 0–6) in 43 cases; (*p* = 0.077)
Montgomery et al. [38]	Patients who received thrombolytic (drug unspecified) for treatment of acute ischaemic stroke.	To assess the impact of ED pharmacists on thrombolytic administration times.	**Retrospective cohort study** conducted in a US Hospital with data collection covering a period of 2 years and 9 months. Participants were assigned to a pharmacist present or pharmacist absent group for analysis.	Reviewing contraindications and co-ordinating administration of thrombolytics including dosing and reconstitution.	97 patient records were included with 38 allocated to the pharmacist present group and 59 to the pharmacist absent group. **Door-to-needle times:** Pharmacist present mean 54 min vs. pharmacist absent mean 74 min; (***p* = 0.004**) <60 min achieved in 27 of 38 cases (71.1%) when pharmacist present vs. 23 of 59 cases (40.0%) in the pharmacist absent group (***p* = 0.002**). <45 min achieved in 16 of 38 cases (42.1%) when pharmacist present vs. 11 of 59 cases (18.6%) in the pharmacist absent group (***p* = 0.012**).
Pandya et al. [35]	Patients who received thrombolytic (rtPA) for treatment of acute ischaemic stroke.	To evaluate and define the pharmacist role on stroke response.	**Retrospective cross-sectional study** of pharmacy resident stroke team pages/calls conducted in a US hospital with data collection covering a period of 12 months.	To respond to emergency stroke team calls and ensure compliance with a site-specific ischaemic stroke acute treatment protocol for blood pressure management, thrombolytic dosing, preparation, and monitoring. Provided as part of an on-call residency; also drug information and clinical pharmacokinetics services.	Of 256 stroke team calls, 46 patients received thrombolytic (rtPA). Of these, thrombolytic had been administered external to the study hospital in 22 cases. For the 24 cases treated at the study hospital there were no deviations from protocol meaning all patients were given thrombolytic within 3 h of symptom onset and there was no thrombolytic drug waste. No door to needle times reported. Three medication errors (including one thrombolytic dosing error) were identified in patients administered thrombolytic external to the study hospital.
Rech et al. [42]	Patients aged ≥18 years who received thrombolytic (rtPA) for acute ischaemic stroke in 4.5 h of stroke symptom onset.	To determine whether pharmacist intervention at bedside in acute ischaemic stroke can reduce door to needle times for thrombolytic treatment.	**Retrospective Cohort study** conducted in a US Hospital with data collection covering a period of 4 years. Participants were assigned to a pharmacist present or pharmacist absent group for analysis.	Evaluate rtPA contraindications, elicit and review medical and medication histories, manage blood pressure. Calculate rtPA dose, prepare and administer rtPA and monitor after administration. Counsel patients and/or carers.	125 patient records were included with 45 allocated to the pharmacist present group and 80 to the pharmacist absent group. **Door-to-needle (rtPA) times:** Pharmacist present median 48 min vs. pharmacist absent median 73 min; (***p* < 0.01**) ≤60 min achieved in 32 of 45 cases (71.1%) when pharmacist present vs. 23 of 80 cases (28.8%) in the pharmacist absent group (***p* < 0.01**)
Roman et al. [44]	Stroke patients who received rtPA.	To assess the impact of EM pharmacists on thrombolytic administration times.	**Pre/post-implementation cohort study** conducted at an Australian hospital with data collection retrospectively and prospectively covering a pre period of 2 years and 7 months and post period of 3 years and 11 years.	Elicit and review medication history, manage acute blood pressure, calculate rtPA dose, prepare and administer rtPA and monitor after administration.	**Door-to-needle (rtPA) times:** Pharmacist present median 61 min vs. pharmacist absent median 73 min; (***p* < 0.012**) <60 min achieved in 59 of 122 cases (48.4%) when pharmacist present vs. 22 of 64 (34.4%) in the pharmacist absent group (*p* < 0.068) **Median LOS in hospital** Pharmacist present median 4.2 days (SD 2.9–7.3) vs. pharmacist absent median 6.6 days (SD 4.1–10.8); (*p* = 0.003) **Mortality in hospital** 24 of 122 cases (19.7%) when pharmacist present vs. 12 of 64 (18.8%) in the pharmacist absent group (*p* = 0.97).
Tsai et al. [40]	Patients who received thrombolytic (drug unspecified) for treatment of acute ischaemic stroke.	To evaluate the quality of patient care and cost avoidance when pharmacists are involved in a multidisciplinary ischaemic stroke team.	**Retrospective cohort study** conducted in a Taiwan hospital with data collection covering a period of 1 year and 10 months pre- and post-intervention. **(Only published abstract available)**	Pharmacist participation in medical rounds.	648 patient records were examined. The number of records pre and post intervention were not reported. Pharmaceutical care quality measured against five measures as set out by the American Heart Association and American Stroke Association Get with the Guidelines—Stroke Program. Intravenous thrombolytic (within 3 h of symptom onset), 24.0% with pharmacist involvement vs. 0.0% without. Early and discharge antithrombotic, 95.9% with pharmacist involvement vs. 93.3% without. Anticoagulation therapy for AF, 71.4% with pharmacist involvement vs. 20.0% without. No change in lipid-lowering medication prescribing rates (51.7% vs. 52.7%). Cost avoidance due to pharmacist involvement was estimated at $2,207,816 NTD.

AF (Atrial Fibrillation); ED (Emergency department); EM (Emergency Medicine); INR (International Normalized Ratio); IQR (interquartile range); LOS (length of stay); NTD (New Taiwan Dollars); rtPA (recombinant tissue plasminogen activator); US (United States).

**Table 7 healthcare-10-02315-t007:** Secondary Care interventions for long-term management post the acute phase of care.

Author(s) and Year of Publication	Target Population	Study Aim(s)	Study Design & Setting	Description of Pharmacist/Pharmacy Student Intervention	Results/Outcomes
Andres et al. [45]	Patients with a stroke or TIA (*n* = 455)	To determine if patients receiving care from the SPC have better outcomes than patients who received UC. To evaluate the total change in BP, LDL, and HbA1c from the time of stroke/TIA to most recent value post SPC intervention.	**Retrospective cohort study** conducted in a US Hospital over 4 months.	Pharmacotherapy intervention: medication review and education.	The **composite end point of hospital readmissions** for stroke/TIA, MI, or new or incidental PAD in the SPC group (*n* = 257) attained. Statistical significance (***p* = 0.013**) when compared to the control group (*n* = 198). Patients who visited the SPC had 4% fewer hospital admissions for stroke/TIA (***p* = 0.125**). All surrogate markers, including blood pressure, Low Density Lipoprotein, Haemoglobin A1c, and smoking status, improved in the SPC group.
Chiu et al. [21]	Patients with ischemic stroke outpatients who visited clinics regularly after stroke for more than 12 months. (*n* = 160)	To assess the management of modifiable risk factors (MRF) adequacy in IS outpatients’ group. To evaluate the importance of pharmacist intervention in a randomized controlled study in hospital.	**Randomized controlled study (RCT)** conducted in a Taiwan hospital, over 6 months.	Educational intervention programme over 6 months regarding side effects, drug interactions, identifying and solving DRPs, and a medication review.	**Differences in lipid profiles, blood glucose, and blood pressure before and after the study.** **BP control change** Intervention 83% Control 43% **(*p* = 0.00)** **Lipid normal level** Intervention 40% Control 27% **(*p* = 0.16)** **Glucose control** Intervention 35% Control 46%. **(*p*= 0.40)**
Greger et al. [49]	Post-stroke or post-TIA patients. (*n* = 342)	To determine pharmacist interventions along with anti-platelet medication monitoring when compared to usual care.	**Retrospective matched (*n* = 171 for each group) cohort study** conducted in a US single centre, outpatient neurology practice.	Medication review, reconciliation, adherence counselling, and risk factors modifications.	**Responsiveness to antiplatelet medication after Pharmacist interventions.** Patients’ responsiveness was 83% at pharmacist intervention group **(*p* < 0.0001)** **Selected interventions frequency in pharmacist group compared to usual care group.** Drug-drug interactions (***p* < 0.0001**) and counselling on adherence (***p* < 0.0008**) were identified in pharmacist group compared to usual care.
Hedegaard et al. [22]	Patients with acute first-time ischaemic stroke, aged ≥18 years and in an emergency ward or neurology department. (*n* = 200)	To assess the multifaceted pharmacist intervention in improving medication adherence for secondary stroke prevention.	**Randomized controlled study (RCT)** conducted in a Denmark hospital over 6 months.	Medication review, an interview, consultation, and three follow-ups via telephone calls.	**MPR of antiplatelets, anticoagulants, and statin one year after discharge.** Median MPRs after 12 months MMPRs (IQR) were 0.95 (0.77–1) in the intervention group and 0.91 (0.83–0.99) in the control group. MPR reduction (3 to 12 months). 5% and 9% in the intervention and control groups, respectively **(*p* < 0.05)**.
Hohmann et al. [26]	Patients diagnosed with TIA or ischemic stroke. (*n* = 255)	To assess if the pharmaceutical care increases the patient’s health related quality of life (HQL).	**A cluster cohort study** conducted in a German rehabilitation hospital and community pharmacies over 12 months.	Counselling interview about medications, mainly secondary preventions regarding side effects, drug interactions, DRP identification and resolution, and a review of medication.	**Patient’s HRQoL** Significant decrease observed in 7/8 subscales at CG. Vitality subscale was significantly decreased in CG than IG **(*p* = 0.027)** **Secondary prevention** 85.3% (IG) and 86.3% (CG) of patients were prescribed with antiplatelet and oral anticoagulant medications accordant with DGN and DSG guidelines. **Patients’ satisfaction with pharmacist interventions** IG was more satisfied with pharmacist interventions compared to CG **(*p* < 0.016)**.
Hohmann et al. [25]	Patients with TIA or ischemic stroke with a Barthel index of over 30 points at discharge time. (*n* = 255)	To assess the impact of of pharmaceutical care on HRQoL by using a SF-36.	**A cluster cohort study** conducted in a German rehabilitation hospital and community pharmacies, over 12 months.	Counselling, medication review, and interviews regarding secondary preventive medications.	**HRQoL** VT subscale was significantly decreased in CG than IG **(*p* = 0.027)** Bodily pain significantly dropped in IG **(*p* = 0.0001)** A significant decrease between the PCS and MCS in CG **(PCS: *p* = 0.023; MCS: *p* = 0.001)** **No statistically significant between the PCS and MCS in IG.** A significant drop in the HRQoL was noticed in 7/8 (RP, BP, GH, VT, SF, RE, and MH) subscales in CG **(*p* < 0.05; *p* < 0.01; *p* < 0.001)**.
Hohmann et al. [34]	Patients with TIA or IS aged ≥18 years and were taking two or more medications during admission and discharge. (*n* = 156)	To explore the frequency and type of DRPs over the pharmaceutical interventions to detect them with TIA patients or IS from admission to discharge from hospital.	**A cross sectional study** conducted in a German hospital, over 6 months.	Medication reconciliation on admission and at discharge, providing information about medication modifications during the period of hospital admission, and reasons for antithrombotic therapy changes, ward rounds participation and detecting DRP.	**Percentage of patients who had a DRA** 271 DRPs happened in 105 out of 155 (67.7%) patients. **Percentage of physician’s acceptance for pharmacist interventions** 89% of pharmacist interventions were accepted by GPs.
Hohmann et al. [27]	Patients with TIA or IS aged ≥18 years and were taking two or more medications during admission and at discharge. (*n* = 312)	To evaluate the adherence of primary care clinicians to prescribing the medication regimes started at the time of in-hospital stroke treatment following pharmacist intervention to improve discharge letter communication.	**Non-randomized trial** conducted in a German hospital over 6 months.	Listing the drugs at admission and discharge in the discharge letter, 3 follow-up calls, providing information in details about the change of drugs during hospital admission, modifications of drugs.	**Overall adherence to pharmacist recommendations by PCPs** IG + 7.6% (90.9% for IG—83.3% for CG) **(*p* = 0.01)** **Antithrombotic drugs adherence to pharmacist recommendations by PCPs** IG + 8.1% (91.9% for IG—83.3% for CG) **(*p* = 0.033)** **Statin adherence to pharmacist recommendations by PCPs** IG + 17.9% (87.7% for IG—69.8% for CG) **(*p* < 0.001)**.
Khalil et al. [51]	Patients who diagnosed with IS, and aged ≥18 years. (*n* = 124)	To examine the management of pharmacological treatment of stroke patients and measure the adherence to stroke management guidelines with and without pharmacist intervention.	**Retrospective audit** conducted in an Australian hospital, over 5 months.	Medication reconciliation and review.	**Percentage of discharged patients on antihypertensive, lipid lowering medications, and antithrombotic.** **Antihypertensive prescribed for secondary prevention:** 83% of patients who were reviewed by pharmacists 59% patients who were not seen by pharmacists **(*p* = 0.005)**. **Lipid lowering agents:** 68% of patients who were reviewed by pharmacists 66% patients who were seen by pharmacists. **(*p* = 0.849)**. **Antithrombotic agents:** 92% of patients who were reviewed by pharmacists 77% patients who were not seen by pharmacists. (***p* = 0.025**).
Lindblad and Howorko [31]	Patients who were recruited with TIA or IS at discharge and with outpatients.	To identify the number and type of pharmacist interventions.	**Cross sectional study** conducted in a Canadian hospital over 6 months.	Medication counselling, a review of medical records, followed-up calls and medication reconciliation.	**Average number of interventions per patient encounter.** 2.8 interventions per encounter. **Average number of patient outcomes associated with pharmacist’s interventions.** 1.9 outcomes per intervention. **Proportion of accepted and rejected pharmacist interventions by prescriber.** Accepted 63.9%
McAlister et al. [24]	Patients diagnosed with confirmed TIA or IS and who had hypertension and dyslipidaemia. (*n* = 275)	To assess the impact of two types of case management (pharmacist-led and nurse-led groups) on global risk of vascular.	**Randomized controlled study (RCT)** conducted in Canada community-dwelling adults, over 6 and 12 months.	Providing lifestyle advice, initiating, and adjusting antihypertensive and lipid-lowering therapy, and discussing risk factor assessments with primary care physicians.	**FRS estimated 10-year risk** At 6 months: median 4.8% (IQR 0.3–11.3%) for the pharmacist arm vs. 5.1% (IQR 1.9–12.5%) for the nurse arm **(*p* = 0.44)**. At 12 months: median 6.4% (1.2–11.6%) vs. 5.5% (2.0–12.0%) **(*p* = 0.83)**. **CDLEM estimated 10-year risk** At 6 months: median 10.0% (0.1–31.6%) vs. 12.5% (2.1–30.5%) **(*p* = 0.37)**. At 12 months: median 8.4% (0.1–28.3%) vs. 13.1% (1.6–31.6%) **(*p*= 0.20)**. **Percentage of participants at six months who obtained normal BP (SBP ≤ 40 mm and fasting LDL ≤ 2.0 mm).** Pharmacist arm 53.1% vs. nurse arm 31.3% achieved the goals of controlling (SBP) and (LDL) **(*p* = 0.005)**.
Nathans et al. [39]	Patients with a stroke or TIA (*n* = 94)	To determine the effect of a pharmacist TOC on hospital readmissions, ED visits, and recurrent events.	**Retrospective matched cohort Study** conducted in a US university hospital over 18 months.	Adjustment of the medication dose. Therapy addition or discontinuation. Monitoring and requesting laboratory tests. Counselling.	**Primary endpoint was 30-day hospital readmissions rate.** No significant difference was found in 30-day readmissions. **(*p* = 0.12)**. **Secondary endpoints included 90-day readmissions, 30 and 90-day emergency department visits, and recurrent stroke rates.** Significant difference found in 90-day readmissions (5.3% vs. 21.3%). **(*p* = 0.001)**.
Nguyen et al. [23]	Patients with stroke history. (*n* = 30)	To assess whether a clinical pharmacist intervention could improve adherence to stroke medications and achieve prevention of stroke goals.	**Randomized controlled study (RCT)** conducted in a US hospital over 6 months. **(Only published abstract available)**	Telephone follow-up calls at 3 and 6 months which include medication adherence evaluation, stroke education, stroke prevention goals reassessment.	**Medication Adherence.** Pharmacists’ intervention group 56% vs. Usual care group 36% Adherence to antithrombotic only (73% vs. 57%). **Achieving the stroke prevention goals.** BP (73% vs. 57%) LDL-C goals (75% vs. 50%) Blood glucose control (75% vs. 50%).

BP (Bodily pain); CDLEM (Cardiovascular disease life expectancy model); CG (Control group); DRPs (Drug-related problems); ED (Emergency department); FRS (Framingham risk score); GH (General health); HRQoL (Health-related quality of life); IG (Intervention group); IQR (Interquartile range); IS (Ischaemic stroke); LDL-C (Low-density lipoprotein—cholesterol); (MCS (Mental component summary); MH (Mental health); MI (Myocardial infarction); MPR (Medication possession ratio); PAD (Peripheral artery disease); PCPs (Primary care physicians); PCS (Physical component summary); RE (Emotional role); RP (Physical role); SBP (Systolic blood pressure); SF (Social functioning); SF-36 (Short form 36); SPC (Pharmacist-run stroke prevention clinic); TIA (Transient ischemic attack); TOC (Transition of care); TWD (New Taiwan dollar); VT (Vitality).

### 3.1. Primary Prevention Interventions

Of the seven primary prevention intervention studies included, two were focused on patient education. One provided a blood glucose and blood pressure screening service with a 10-min educational session (the Act FAST educational intervention) delivered by student pharmacists to raise community awareness of the signs and symptoms of stroke [30]. In the other study, pharmacists provided education for hospital outpatients with atrial fibrillation (AF) about direct oral anticoagulant (DOAC) therapy including the importance of adherence [36]. All the remaining primary prevention studies centered on patients with either diagnosed or undiagnosed AF. Two studies showed an improvement in the appropriate use of antithrombotics in patients diagnosed with AF by hospital-based pharmacist assessment of patient stroke risk, followed by recommendations for therapy optimisation [28,29]. The positive impact of hospital pharmacist optimisation of oral anticoagulation for patients with diagnosed AF was also demonstrated by clinical audit [50]. Two further studies assessed a pharmacist screening intervention for undiagnosed AF (e.g., pulse palpation, ECG recording and interpretation) in the community pharmacy setting [32,33]. In both studies, pharmacy customers with newly identified suspected AF, and therefore elevated stroke risk, were referred to their general practitioner (GP) for further management. Of 1000 customers screened who were over 65 years old, 1.5% (95% CI, 0.8–2.5%) had previously undetected suspected AF [32]. Targeted screening of high-risk individuals identified 28.6% (*n* = 35) with abnormal rhythms [33].

### 3.2. Secondary Prevention Interventions

Most studies (24 of 31) investigated the impact of pharmacist interventions for the secondary prevention of stroke. These interventions were at different stages of the patient treatment pathway and ranged from the immediate emergency treatment of stroke (i.e., at initial emergency department presentation) [35,37,38,40,41,42,43,44,46,47,48] and intermediate hospital-based post-acute phase care [22,27,34,51] through to long-term follow-up post-stroke in outpatient clinics [23,31,39,45,49], tertiary referral centres [21] or community care [24,25,26].

#### 3.2.1. Emergency and Acute Care Interventions

Pharmacist interventions in the emergency and acute care of patients diagnosed with stroke were described in eleven studies [35,37,38,40,41,42,43,44,46,47,48]. These were categorised as secondary prevention because appropriate acute treatment has the potential to reduce and prevent stroke recurrence. This acute phase of care is usually considered to have ended either at the time of acute stroke unit discharge or by 30 days of hospital admission.

Nine studies [35,37,38,41,42,43,44,46,47] reported the impact of pharmacist involvement in emergency stroke response teams responsible for the preparation and administration of emergency thrombolysis (e.g., intravenous recombinant tissue plasminogen activator (rtPA); alteplase). Emergency thrombolytics are high-cost drugs and involve complex patient assessment criteria and intense patient monitoring. Their use is associated with a high risk of bleeding and the potential for significant medication error. For patients diagnosed with ischaemic stroke, the potential beneficial effects of thrombolysis are time dependent, and it is well-established that reducing time to treatment from presentation (i.e., the door-to-needle time, DTN) results in better neurological outcomes, reduced adverse effects and improved mortality [52,53,54,55,56,57]. In the United States, national quality initiatives target a DTN of 60 min [58]. Meta-analysis of seven studies revealed that the odds of DTN ≤45 min were 2.69 ([95% CI: 1.95–3.72]; *p* > 0.001) times higher with pharmacist involvement than without (Figure 2). As well as reducing DTN times, pharmacists are reported to streamline thrombolytic preparation and administration processes [41] and minimize protocol deviations [35]. In addition, pharmacists’ involvement in emergency and acute care (i.e., 10 days after hospital admission with stroke) is also reported to improve anticoagulation therapy and generate substantial savings in healthcare costs during inpatient stays [40]. Similarly, pharmacists’ reconciliation of medicines and pharmaceutical interventions at hospital admission, and during inpatient stays to the point of discharge, have been shown to resolve drug related problems, optimise pharmaceutical care [34] and improve adherence to prescribing guidelines for secondary prevention [51]. Pharmacists are not only reported to have a role in inpatient care but also at the transition of care. One study demonstrated that detailed input into discharge letters by pharmacists ensured more patients were maintained on optimised therapy by their general practitioners. Results were significantly better for secondary preventative medicines such as anticoagulants (83.8% control group vs. 91.9% intervention group [*p* = 0.033]) and statins (69.8% control group vs. 87.7% intervention group [*p* < 0.001]) [27].

#### 3.2.2. Long-Term Care Interventions

Ten studies [21,22,23,24,25,26,31,39,45,49] described a broad range of long-term interventions by pharmacists post the acute phase of stroke treatment. All were conducted in the outpatient setting with one using community-based pharmacies and publishing results as part of two articles [25,26].

All the pharmacist interventions reported were multicomponent and ranged in duration from a one-off consultation [39], a specified number of consultations and follow-up over 6 [21,22,23,24,31] or 12 months [25,26], to a consultation with follow-up according to patient need [45,49] All, except one which involved only telephone calls at 3 and 6 months [23], employed face-to-face patient-pharmacist interaction as part of the intervention. Common to all interventions was counselling or patient education on modifiable risk factors (including lifestyle advice) and, except for one educational intervention [21], all included some form of medicines optimisation activity by the pharmacist, either alone or through recommendations to prescribing clinicians/physicians.

### 3.3. Common Outcome Measures across All Interventions

Across all studies, six common outcomes were identified; DTN for acute thrombolysis (as reported in the meta-analysis above); medicines adherence; medicines optimisation; clinical risk factor modification; and patient-related outcomes such as health-related quality of life (HRQoL).

#### 3.3.1. Medicines Adherence

Three studies measured the impact of the pharmacist’s intervention on patients’ adherence to stroke medication [22,23,36]. One RCT, which was published as an abstract only, reported higher medication adherence at 6 months after discharge and a telephone intervention by a pharmacist when compared to control: 56% vs. 36% for all medication and 100% vs. 88% for antithrombotics in 30 patients [23]. However, no definition of the adherence measure was supplied. In contrast, in a study of pharmacist medication review, interview and 3 follow-up telephone calls within 6 months of discharge, the adherence of 90 patients to antiplatelet therapy, anticoagulants and statins post-hospitalisation was not statistically different between intervention and control (composite Medication Possession Ratio (MPR) over one year: median MPRs (IQR) 0.95 (0.77–1) vs. 0.91 (0.83–0.99)) [22]. Adherence for both groups was high throughout the study with a small, statistically significant, decrease over time (intervention group 5% MPR reduction (*p* < 0.05) vs. control 9% MPR reduction (*p* < 0.05)) [22]. Lee et al., also measured adherence using MPR. Twenty patients in this study, newly started on dabigatran therapy for atrial fibrillation, attended an outpatient clinic with pharmacist monitoring and education provision over the first three months of treatment. Compared to historical controls, MPRs at three months were slightly higher but the difference was not statistically significant (93.1% vs. 88.3%) (*p* = 0.16) [36].

#### 3.3.2. Medicines Optimisation

Pharmacist interventions were identified as medicines optimisation where pharmaceutical care included medicines review, medicines reconciliation, identification and resolution of drug related problems. To allow comparison of similar outcome measures for different treatment regimens, findings have been separated into primary and secondary prevention optimisation.

##### Medicines Optimisation for Primary Prevention of Stroke

Three primary prevention studies [28,29,50] investigated the impact of anticoagulant and antithrombotic medicines optimisation for patients at risk of stroke. In the hospital setting, Bajorek et al. reported that for patients admitted at-risk of stroke (e.g., with AF), a pharmacist intervention identified 78 of 218 (35.8%) required changes to optimise existing antithrombotic therapy according to locally developed evidence-based guidelines. More effective therapy to reduce stroke risk (e.g., prescription or change of preventative medication) was required in 60 (76.9%) cases and the remainder required a change to potentially less effective but safer options [28]. Similarly, a hospital-based, pharmacist-led stroke risk assessment programme for 134 inpatients with AF resulted in an increase in warfarin use from 74% on admission to 98% at discharge, and 50 recommendations for therapy change of which 44 (80%) were agreed and implemented. More effective therapy was required in 30 of the 44 (68%) cases [29]. Clinical audit also demonstrated a pharmacist’s impact in a general practice setting, where 77% of 382 anticoagulant and antithrombotic optimisation recommendations for patients with AF were agreed by general practitioners [50].

##### Medicines Optimisation for Secondary Prevention of Stroke

Of the three [27,34,51] secondary prevention pharmacist interventions conducted during admission and at the discharge phase of stroke treatment, two reported medicines optimisation during the patient’s hospital stay [34,51]. Sub-analysis of data from Khalil et al.’s hospital-based audit of 56 patient medical records against national guidelines, demonstrated a statistically significant increase in the percentage of patients discharged on appropriate secondary preventative antihypertensive agents (83% vs. 59%, *p* = 0.005) and antithrombotic medicines (92% vs. 77%, *p* = 0.025) [51]. Hohmann et al. investigated the identification and resolution of drug related problems (DRPs) with medicines reconciliation on admission and regular medicines review by pharmacists. On average, pharmacists identified 1.8 ± 2.0 DRPs per patient (*n* = 156) with 89% of intervention recommendations adopted by clinicians. Importantly, a fifth (20.7%) of DRPs related to the indication for a prescribed drug and included drugs not prescribed which should have been (i.e., missing drugs). Most of the missing drugs (80% of 36) were stroke-related secondary preventative medicines [34].

All ten studies involving long-term care and secondary prevention in the outpatient setting used medicines optimisation with five reporting the numbers of pharmacist interventions made in response to DRP identification [26,31,39,45,49]. Interventions recorded included medicines counselling, monitoring (e.g., platelet function for responsiveness to antiplatelet therapy [49]), discontinuation of medicines (e.g., inappropriate duplication of secondary prevention [26]), adding a therapy, dose titration, dose changes, formulation changes, switching a therapy, preventing or resolving adverse drug reactions and rectifying suboptimal dosing (either too high or too low). The average number of interventions per patient encounter was reported in two studies as 3.5 [39] and 2.8 [31]. One of these studies also gave a measure of pharmacist intervention acceptance by prescribers which was not identified in the other studies (*n* = 432, 63.9% interventions accepted; 8.3% rejected; 16.2% prescriber acceptance not required; 11.6% unrecorded) [31].

#### 3.3.3. Modification of Clinical Risk Factors

While risk factor modification potentially includes lifestyle changes as well as changes in disease-related clinical measures, only two of the included studies reported measures of lifestyle change: smoking status [24,45], physical activity [24], and weight [24]. Four studies investigated the impact of pharmacist intervention on clinical risk factor-related outcome measures for the secondary prevention of stroke (e.g., blood pressure, lipid levels, blood glucose) [21,23,24,45].

##### Blood Pressure

Three of the four studies reported improvement in hypertension management and attainment of blood pressure (BP) goals for pharmacist interventions when compared to control. Only one showed a statistically significant reduction whereby, at the end of a 6-month intervention of monthly educational sessions, 83.3% (*n* = 78) adequate hypertension management with intervention was attained as compared to control 43.4% (*n* = 76), *p* < 0.001 [21]. Nguyen et al., reported 73% vs. 57% attainment of BP goals for telephone intervention (total sample size 30 patients; sample size per group not reported) [23]. In contrast, Andres et al., reported higher BP goal attainment in the control group for their stroke clinic intervention (60.8%, *n* = 245 vs. 66.7%, *n* = 165). However, the control group had a lower baseline incidence of hypertension at the time of stroke when compared to the intervention group (83.8%, *n* = 166 vs. 92.9%, *n* = 239), suggesting that goal attainment was more difficult in the intervention group [45].

##### Lipid and Blood Glucose Control

Lipid and blood glucose control results were less comparable across the four studies. All reported a positive impact of pharmacist intervention. Two studies found a greater attainment of lipid and blood glucose goals when compared to control [21,23]. Of these, Nguyen et al. reported that 80% of the intervention group (total sample size 30 patients; sample size of the intervention group not reported) achieved lipid and blood glucose goals compared to about half in the control group [23]. Similarly, greater reductions in lipid and blood glucose levels were reported by Andres et al., for their intervention group over control (HbA1c 0.6% reduction vs. control at 0.1%; average low-density lipoprotein (LDL) decrease 23 mg/dL vs. control 9 mg/dL) [45]. However, only one study reported statistical testing, with no significant differences found between intervention and control groups [21]. For systolic BP and LDL, monthly intervention by a prescribing pharmacist over 6 months resulted in greater attainment of targets at the end of the study when compared to non-prescribing nurse-led management, 53.1% (*n* = 81) vs. 31.3% (*n* = 83), *p* = 0.005 [24].

#### 3.3.4. Clinical Outcomes

For the primary prevention of stroke, only two studies investigated clinical outcomes which were not surrogate markers [28,36]. Both involved the management of anticoagulation by pharmacists and suggest that pharmacist-managed clinics are comparable to pivotal studies or control for stroke risk and bleeding rates [28,36].

For the secondary prevention of stroke, three studies reported the impact of pharmacist intervention on hospital readmission [22,39,45]. While one study reported no significant difference to control for a 6-month follow-up, pharmacist-led, multicomponent intervention [22], the other two studies reported statistically significant differences in favour of pharmacist involvement in stroke clinics [39,45]. Readmission for stroke, myocardial infarction (MI) and peripheral arterial disease (PAD) was 9.3% (*n* = 24) vs. 17.2% (*n* = 34) *p* = 0.013 [45] and 90-day readmission rates 5.3% (5/94) vs. 21.3% (20/94), *p* = 0.001 [39].

#### 3.3.5. Patient Outcomes

Only secondary prevention studies investigated the impact of a pharmacist intervention on a patient’s quality of life or satisfaction with care. Lindblad and Howorko reported that of 834 interventions conducted by pharmacists, 286 (34.3%) were anticipated to improve physical, mental, or social function or satisfaction with care [31]. Another study found that pharmacist intervention gave approximately one third of patients more confidence and better skills to use medicines correctly [22]. However, the most comprehensive evaluation was provided by a study of 255 patients with outcomes reported in two separate articles [25,26]. HRQoL at the beginning of the study was similar between pharmacist intervention and control groups. After 12 months and without significant changes in patient health status for either group, the reported HRQoL had declined significantly in 7 of 8 variables for the control group compared to a statistically significant decline in only one variable (bodily pain) for the intervention group [25,26]. In addition, the intervention group experienced significantly greater satisfaction than control and rated the intensive counselling from pharmacists as highly beneficial and informative [26].

## 4. Discussion

This systematic review provides evidence of a range of pharmacist interventions in the provision of pharmaceutical care for both the primary and secondary prevention of stroke in various settings, from community pharmacies’ role in improving population awareness regarding modifiable risk factors for primary prevention to pharmacists’ contributions to acute, intermediate, and long-term management for the secondary prevention of stroke. Most studies employed only quantitative methodology and qualitative exploration of the pharmacists’ role in the primary and secondary prevention of stroke was lacking. Three of the seven primary prevention studies and thirteen of the twenty-four secondary prevention studies showed a statistically significant and positive impact of the pharmacist’s role. These findings align with those of previous reviews evaluating the pharmacist impact on the management of stroke in different work settings [59].

### 4.1. Primary Prevention

The value of community pharmacist interventions in delivery of public health education and screening for major disease is well documented [60,61]. This review demonstrates the potential that pharmacists and pharmacies have in reducing stroke burden with primary prevention through education and screening. Several countries employ community pharmacy education programmes to target high risk individuals [62,63]. Such education is effective in improving patient knowledge of cardiovascular disease [64], modifying behaviours, and decreasing stroke incidence [56,63,65]. The pharmacist’s role in primary prevention of stroke extends from community pharmacy into general practice and outpatient clinics. This review shows how pharmacist input into general practice caseloads can lead to optimisation of anticoagulation for high-risk individuals [50]. As well as identifying risk factors for the primary prevention of stroke, pharmacist management of anticoagulant medicines for primary prevention is comparable to usual care [28,36]. While pharmacist-led anticoagulation management (in this review) shows no difference in the adverse events experienced with ‘usual care’, older studies report a reduction in thromboembolic events or rates of bleeding in pharmacist-led anticoagulation clinics [66,67,68]. Several studies have shown high agreement between pharmacists providing medicines optimisation recommendations and clinicians [69,70,71]. The high acceptance rates of pharmacist treatment optimisation recommendations by prescribing clinicians in this review are a positive indicator of the value of pharmacist input and evidence to support the evolving and increasing prescribing role of pharmacists in the management and primary prevention of stroke. In this way, pharmacists can flexibly support the healthcare workforce to deliver care especially, for example in the UK, with transitions in pharmacy education towards independent prescribing at registration [72].

### 4.2. Secondary Prevention Emergency Treatment

Acute stroke treatment includes urgent administration of thrombolytic agents to minimize the risk of disability and death [73]. Several guidelines advocate thrombolytic therapy within 4.5 h of symptom emergence, with a door-to-needle (DTN) target time of 60 min [74,75]. Importantly, the meta-analysis in this systematic review provides strong evidence that where a pharmacist was integrated into dedicated stroke treatment teams, DTN times were significantly reduced. The long-term benefits of speedy thrombolytic therapy are numerous to both patient and the wider health system in terms of outcomes and cost. The role of the pharmacist in this setting is reported to involve evaluation of patient eligibility, suitability of thrombolytic orders and enhancing order accuracy [38,42]. More generally, pharmacists working within emergency departments (ED) are known to add value through medication error identification, drug therapy optimisation, medication use enhancement and guidelines adherence improvement [76,77,78,79]. This systematic review adds further evidence to demonstrate the benefit of pharmacist input into the acute care of patients with suspected or confirmed stroke as part of a multidisciplinary specialist team.

### 4.3. Secondary Prevention Long-Term Management

Recent meta-analyses provide strong evidence for the positive impact of pharmacist interventions on cardiovascular risk factors and outcomes [80]. The evidence for secondary prevention of stroke from this systematic review is weaker due to the heterogeneity of the studies. However, the studies in this review show that pharmacists can have a broad-ranging and beneficial role in case management post-discharge. Through multicomponent interventions, all of which included medicines optimisation, pharmacist input in this review had the potential to improve pharmaceutical care (e.g., prevention of adverse drug reactions) and clinical outcomes (e.g., risk factor modification such as blood pressure control, cholesterol, blood glucose and glycated haemoglobin (HbA1c)). With an improvement in modifiable risk factors there is also the potential to reduce re-hospitalisation, mortality rates and economic burden. Few of the studies in this review analysed the impact of pharmacists as prescribers. As a result, the impact of pharmacist interventions in the prevention of stroke were dependent on successful communication and acceptance of recommendations by prescribers. In the hospital setting, where the multidisciplinary team works together in close proximity, communication may be easier. However, studies have shown that communication between community pharmacists and prescribers in the management of stroke needs to be improved to deliver optimum treatment [81]. As pharmacists’ roles progress to incorporate prescribing, their impact will need to be reassessed. Like other studies of interventions for patients with chronic conditions (e.g., heart failure, hypertension, asthma) [82,83,84], this systematic review also highlights that pharmacist-led case management has a positive impact on a patients’ health-related quality of life (HRQoL) and satisfaction with pharmaceutical care.

### 4.4. Common Findings for Both Primary and Secondary Prevention of Stroke

In both the primary and secondary prevention of stroke, this systematic review provides minimal evidence for pharmacist interventions in supporting patients to be adherent to their treatment regimens. Only one of the three studies measuring patient adherence to medicines showed the pharmacist intervention to improve patient adherence to medicines in an outpatient setting [23]. However, for all three studies [22,23,36] interventions were diverse, duration of follow-up ranged from 3 to 12 months and only one study had medicines adherence as its primary outcome [22]. Interventions guided by pharmacists have been shown to enhance adherence to medicines and disease control for patients with various conditions [85] particularly cardiovascular disease [86,87,88]. This suggests that further investigation is warranted for the primary and secondary prevention of stroke.

## 5. Limitations of the Review

There are a few noteworthy limitations in this review. The number of studies exploring the pharmacist role in primary and secondary prevention of stroke was low and thus studies reported only as conference abstracts and audits were included. The quality of the abstracts and audits could not be assessed, and information provided within these types of publications was limited. In addition, outcomes from articles describing pharmacist integration in primary and secondary prevention of stroke treatment were often inconsistent across the studies due to the heterogeneity of study design and outcome measures. This meant that conclusions drawn and correlation of outcome measures between studies was limited. Meta-analysis was only achievable for one type of pharmacist intervention in emergency treatment of stroke but could not be considered in the synthesis of data from other interventions due to heterogeneity of study designs, interventions and reported outcomes. In addition, the meta-analysis conducted in this study should be interpreted with caution. The results included are from retrospective cohort studies which may be subject to bias (e.g., environmental or participant factors), which may counteract or enhance the effects of pharmacist involvement. A search for dissertations and unpublished studies was not executed and only studies published in the English language were included. Non-significant or negative research findings may not be published in the current literature due to publication bias and this could lead to an exaggeration of the benefit of the pharmacist’s role in primary prevention and secondary prevention of stroke.

## 6. Conclusions

The role of pharmacists in the primary and secondary prevention of stroke is varied and spans the patient treatment pathway, from public education and screening through to acute management and then long-term follow-up. This review shows the potential for an expanding role in the community provision of public health education and screening services for the prevention of stroke. It also demonstrates the potential benefits of pharmacist management of anticoagulation clinics for people with atrial fibrillation. The meta-analysis in this study shows that the most robust evidence for the positive impact of pharmacist input into the management of stroke is currently found in the emergency care of patients, where pharmacist involvement in hospital stroke management teams significantly reduces the time taken to administer appropriate thrombolytic therapy. The evidence for the long-term management of patients post-stroke is diverse and more studies are needed to be able to strengthen conclusions. However, this review demonstrates that pharmacists, in an outpatient or community setting, can play a crucial role in modifying treatment to manage the clinical risk factors which could contribute to stroke recurrence. By involving pharmacists in one-to-one patient management, they can have a beneficial role in counselling, provision of health education and advice which in turn has the potential to improve health-related quality of life and satisfaction with their healthcare experiences.

## Figures and Tables

**Figure 1 healthcare-10-02315-f001:**
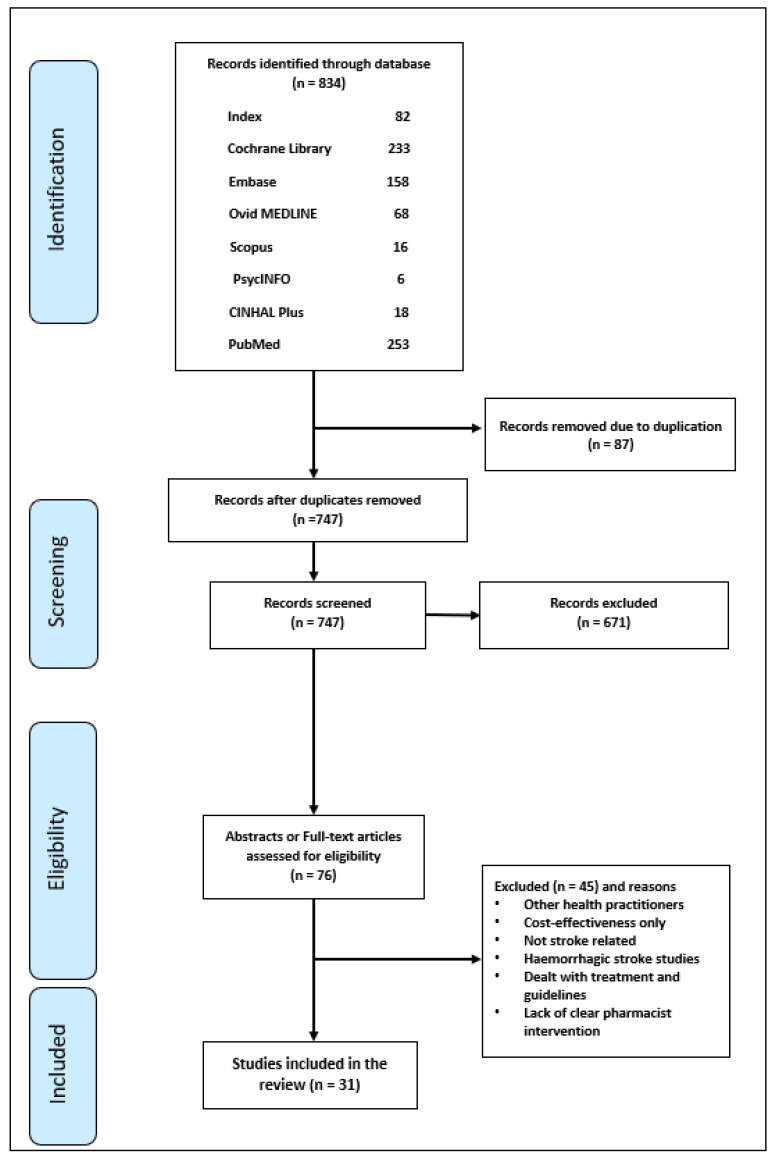
PRISMA flow chart of screening process and reasons for exclusion of studies.

**Figure 2 healthcare-10-02315-f002:**
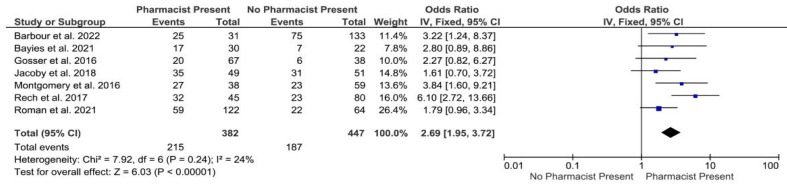
Forest plot illustrating the impact of pharmacist involvement in DTN for thrombolytic therapy where the number of events listed is DTN < 45 min and the total represents the number of cases in each group [37,38,42,43,44,46,47].

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
