# Peer review of "The Role of Pharmacists in Providing Pharmaceutical Care in Primary and Secondary Prevention of Stroke: A Systematic Review and Meta-Analysis"

_healthcare, 2022, doi:10.3390/healthcare10112315_

Round 1
Reviewer 1 Report
Thank you for the opportunity to rev new this manuscript. This is a well conducted study and well written written manuscript. Authors performed a systematic review and meta analysis to report the role of pharmacists in providing care in primary and secondary prevention of stroke. The authors have reported the methods well. The outcomes categories assessed and presented are comprehensive.the results and limitations have been adequately discussed. This study adds significantly to the literature. Only a few minor typo/ grammar/ font size issues were identified in the line numbers mentioned below : 583-586 488-490 351 258-261 154 Figure 1 :the identification box (n=834) is not a sum of the numbers reported for separate database. Please check the numbers
Author Response
Dear Editor
The authors would like to thank the reviewers for their helpful comments. We have outlined the changes made in response to the comments below.
Our Response:
Thank you for this feedback. We have amended the grammar and font sizes in the line numbers listed by the reviewer.
For figure 1, the numbers reported were in the diagram that was submitted. However, converting the diagram to the journal's format led to some numbers being hidden in the text box. This has now been resolved by expanding the size of the text box.
Reviewer 2 Report
I am honored to review this manuscript.
This report discussed the role of pharmacists in primary and secondary prevention of stroke by the systematic review.
The topic is met the journal's scope but I recommend revising the manuscript before publishing on the healthcare journal.
Style: The abstract should be a total of about 200 words maximum. (Please check Instructions for Authors)
Major:
The authors indicated the result of meta-analysis only in Fig. 2. However, they discussed about various topics in the result section, there are no statistical analysis with them. So, we are not sure whether the written results are an important or strong evidence. As described in the limitation section, there may have heterogeneity of study design or research. Even if that were the case, it is necessary to discuss based on the statistical investigations for most important and strong evidence.
Minor:
In figure 1, the number of records identified through data base is described n=843, but a sum of numbers below is not 843. Please explain.
In figure 2, "events" should be better to change to other word. (DTN<45?) Please consider.
The studies in the meta-analysis in Figure 2 are all retrospective cohort studies. A meta-analysis using not RCT study, special care must be taken in interpretation. Please describe clearly.
Author Response
Dear Editor
The authors would like to thank the reviewers for their helpful comments. We have outlined the changes made in response to the comments below.
Our response to your comment 1: Thank you for this comment. The number of words has been reduced to 198.
Our response to your comment 2: Thank you for this feedback. Due to the heterogeneity of study design and outcome measures we were only able to conduct statistical analysis on one outcome measure (i.e., DTN<45 minutes) for which the meta-analysis was conducted. We have explained this in section 2.4 ‘Data analysis’ and in section 5 ‘Limitations of the review’.
In response to the reviewer’s comments, we have amended the wording about the meta-analysis in the discussion section 4.2. We have also added a statement to acknowledge that thew strength of our other findings is weaker than that found by meta-analysis in section 4.3. In addition, we have further refined the conclusion of the article to emphasise the strength of the meta-analysis.
Our Response to your comments 3: The numbers reported were in the diagram that was submitted. However, converting the diagram to the journal's format led to some numbers being hidden in the text box. This has now been resolved by expanding the size of the text box.
Our Response to your comment 4: Thank you for identifying this issue. The meaning of "event" has now been explained and included in the caption of Figure 2.
Our Response to your comment 5: Thank you for this valuable comment. We have now clarified this in section 5 ‘Limitations of the review’.
Reviewer 3 Report
This systematic review and meta-analysis of the role of pharmacists in stroke prevention summarizes the literature on an important topic. The manuscript could be strengthened by addressing the following issues:
Introduction line 93-95: The aims don't include the meta-analysis. This should be added.
Search strategy: The PRISMA guidelines call for the inclusion of the full search strategies for all databases, registers, and websites, including any filters and limits used. This should be included as an appendix.
Quality Assessment line 180: The JBI critical appraisal tools should have a citation.
Table 1: This is not an author issue but the heading of Table 1 cannot be read. The journal should allow for improved Table structures for a systematic review.
Results line 212: 'bar' should be replaced with 'but'
Figure 2: The headings make it difficult to understand. What are the events and total? Are the events positive events (DTN <45 minutes) out of the total cases? This needs to be made more clear.
Discussion lines 513-515: The way this is written it seems that 3 out of 31 studies showed impact for primary prevention and 13 out of 31 for secondary prevention showed impact, when it's really 3 out of 7 and 13 out of 24. I recommend re-writing to make this clear.
Author Response
Dear Editor
The authors would like to thank the reviewers for their helpful comments. We have outlined the changes made in response to the comments below.
Our Response to your comment 1: Thank you for this comment. This has now been included.
Our Response to your comment 2: Thank you for identifying this oversight. The full search strategies have now been included as an appendix and attached in response to this comment.
Our Response to your comment 3: Thank you for identifying this oversight. This issue has now been resolved, and the citation has been added. Due to the inclusion of this reference, all reference numbers from 19 onwards have been amended.
Our Response to your comment 4: Thank you for raising this issue which seems to have occurred in the transfer of table from the original submission to the Journal format. This has now been amended by increasing the row height for the table.
Our Response to your comment 5: This has been replaced as requested.
Our Response to your comment 6: Thank you for identifying this issue. To make the figure clearer, the meaning of "event" and “total” has now been explained and included in the caption of Figure 2.
Our Response to your comment 7: Thank you for this feedback. This has now been re-written as requested.

Round 2
Reviewer 2 Report
I recommend publishing this manuscript as this is.